# HIV envelope V3 region mimic embodies key features of a broadly neutralizing antibody lineage epitope

Daniela Fera [1], Matthew S. Lee[1], Kevin Wiehe[2,3], R. Ryan Meyerhoff[3,4], Alessandro Piai [5], Mattia Bonsignori[2,3], Baptiste Aussedat[6], William E. Walkowicz[6], Therese Ton[7], Jeffrey O. Zhou[8], Samuel Danishefsky[6], Barton F. Haynes[2,3] & Stephen C. Harrison[1,9]

HIV-1 envelope (Env) mimetics are candidate components of prophylactic vaccines and potential therapeutics. Here we use a synthetic V3-glycopeptide ("Man$_9$-V3") for structural studies of an HIV Env third variable loop (V3)-glycan directed, broadly neutralizing antibody (bnAb) lineage ("DH270"), to visualize the epitope on Env and to study how affinity maturation of the lineage proceeded. Unlike many previous V3 mimetics, Man$_9$-V3 encompasses two key features of the V3 region recognized by V3-glycan bnAbs—the conserved GDIR motif and the N332 glycan. In our structure of an antibody fragment of a lineage member, DH270.6, in complex with the V3 glycopeptide, the conformation of the antibody-bound glycopeptide conforms closely to that of the corresponding segment in an intact HIV-1 Env trimer. An additional structure identifies roles for two critical mutations in the development of breadth. The results suggest a strategy for use of a V3 glycopeptide as a vaccine immunogen.

---

[1] Laboratory of Molecular Medicine, Boston Children's Hospital, Harvard Medical School, Boston, MA 02115, USA. [2] Department of Medicine, Duke University School of Medicine, Duke University Medical Center, Durham, NC 27710, USA. [3] Duke Human Vaccine Institute, Durham, NC 27710, USA. [4] Department of Immunology, Duke University School of Medicine, Duke University Medical Center, Durham, NC 27710, USA. [5] Department of Biological Chemistry and Molecular Pharmacology, Harvard Medical School, Boston, MA 02115, USA. [6] Department of Bioorganic Chemistry, Sloan Kettering Institute, New York, NY 10065, USA. [7] Department of Biology, Swarthmore College, Swarthmore, PA 19081, USA. [8] Department of Chemistry and Biochemistry, Swarthmore College, Swarthmore, PA 19081, USA. [9] Howard Hughes Medical Institute, Harvard Medical School, Boston, MA 02115, USA. Correspondence and requests for materials should be addressed to D.F. (email: dfera1@swarthmore.edu)

Mimicry of three-dimensional (3D) protein surfaces is a potentially valuable strategy for probing and modulating protein−protein interactions. Peptide mimetics have been explored for drug development and vaccine research in a wide variety of contexts (reviewed in Gross et al.[1]). The highly immunogenic third variable loop (V3) of the HIV-1 envelope protein (Env) is a plausible target of such efforts[2], but glycosylation at two or more V3-loop positions in the native trimer[3–7] has limited the scope of previous work.

The Env V3 loop, parts of which are less variable than other "variable" loops on the Env gp120 subunit, is almost always ~35 residues long. Its tip bears a conserved [312]-GPGR/Q-[315] sequence motif, and its base includes a conserved [324]-GDIR-[327] motif with an N-linked glycan frequently present at position 332 (HXB2 numbering scheme)[8]. Members of the V3-glycan class of broadly neutralizing antibodies (bnAbs) penetrate the Env "glycan shield" and contact both carbohydrate and protein components at the V3 loop base, which includes the GDIR motif[9]. These bnAbs usually require a glycan at N332 for high affinity, and some of them also require other glycans, such as one at N301. Thus, members of this class of bnAbs can have different modes of Env recognition[9–14]. The potency and breadth of antibodies that bind this site have made their elicitation a high priority for immunogen design[15, 16].

Certain glycans on HIV Env, like most glycoproteins, are heterogeneous and are usually expressed as a population of glycoforms[17] (glycosylation variants). While the N332 and N301 glycans are part of an oligomannose patch on HIV Env[18, 19], the types of glycan processing, or lack thereof, at this site are determined by steric environment. Moreover, complex-type glycans are influenced by the glycosyltransferase activities of the cell in which they are produced[18]. Thus, alterations of HIV Env could alter the glycan types at N332 and N301. Such heterogeneity creates a vaccine-design challenge that extends to any glycosylated immunogen[20]. Chemical synthesis of glycopeptides can in principle overcome this hurdle, by generating glycopeptides with homogeneous glycosylation patterns[21]. Moreover, in the specific case of the HIV-1 Env V3 loop, a synthetic mimic of a "minimal" loop could also present the bnAb epitope without interference from other surface features—e.g., the nearby gp120 V1V2 loop, which can interfere with V3 loop recognition[22, 23].

A synthetic V3-loop glycopeptide ("Man9-V3"), described recently, presents the conformational epitopes needed for the binding of N332 glycan-dependent bnAbs[24, 25]. The peptide design derives from the JRFL mini-V3 gp120 outer domain, used previously for determining a crystal structure with the Fab of bnAb PGT128[9]. The synthetic V3 glycopeptide, which has Man9 glycans ($Man_9GlcNAc_2$) at N332 and N301[24] and a 14-residue tip deleted, was used to identify the V3-glycan bnAb DH270.6 (12.9% mutated nucleotides and 71% neutralization breadth[12]) by labeling the B-cell receptor (BCR) on cells from the HIV-1 infected donor CH848; the antibody is part of a previously described lineage from this individual (Fig. 1a)[12].

We describe here a high-resolution crystal structure of "Man9-V3" bound with the single chain variable fragment (scFv) of the N332-glycan dependent bnAb, DH270.6. We have also determined a structure of the Fab of a clonally related member of this

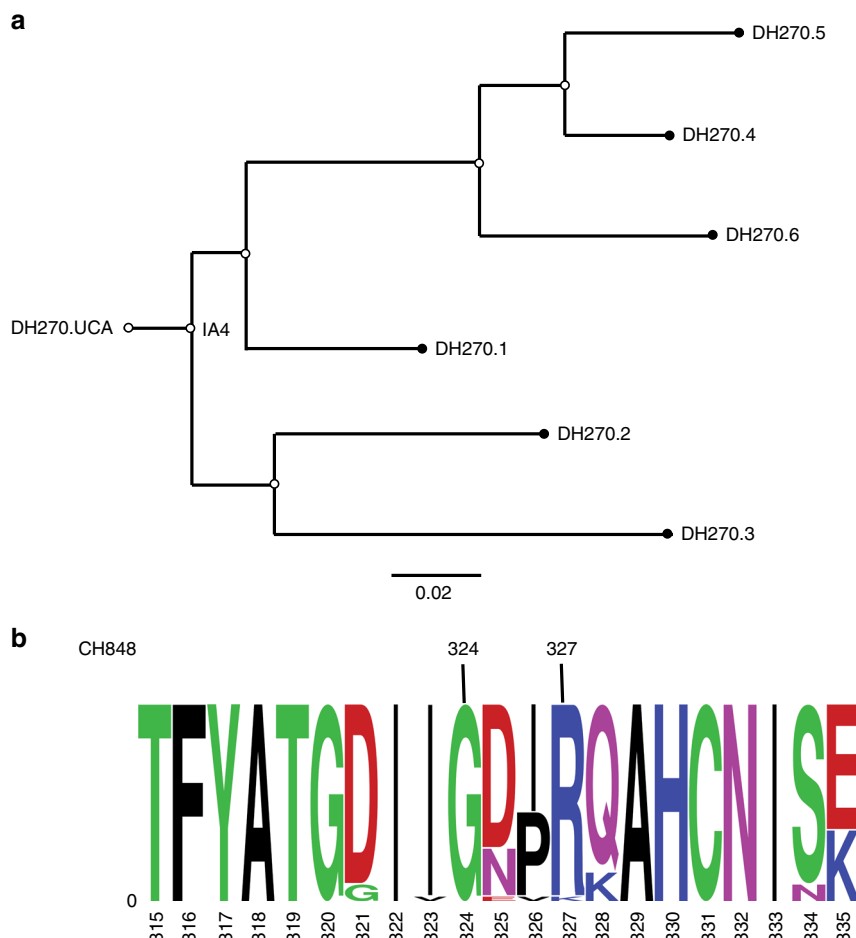

**Fig. 1** Antibody and virus data. **a** DH270 clonal lineage tree. **b** Sequence logo of the V3 region of CH848 autologous viruses. The beginning and end of the GDIR motif are indicated. The frequency of each amino acid at each site is indicated by its height

lineage, DH270.3 (11.8% mutated nucleotides and 42% neutralization breadth), in complex with a synthetic Man$_9$ glycan[26]. These clonally related antibodies have similar numbers of mutation, but their neutralization breadth and potency differ substantially. Through structural analysis of these two related antibodies, we can account for differences in breadth and affinity between the branches of the lineage from which these two antibodies derive and understand the somatic mutation events that led to enhanced binding affinity and neutralization breadth.

Our structures account for the roles of two low probability somatic mutations, introduced at different stages during antibody affinity maturation that correlate with an increase in neutralization breadth. The mutated residues anchor DH270 lineage members onto the V3 loop GDIR region and contact the high mannose moiety of the N332 glycan. These interactions resemble those made by PGT124 and PGT128, which target the same Env epitope[9, 10, 27], but the epitope contacts by DH270-lineage members are with different complementarity determining regions (CDRs) and different amino-acid residues. We further show that the synthetic V3 glycopeptide is a mimic of the V3 region of an intact Env trimer and suggest that it might be useful both as an immunogen and as a probe for determining interactions between other antibodies and HIV Env.

## Results

**Crystal structure of the DH270.6-Man$_9$-V3 complex**. To characterize the binding site (epitope) for DH270 lineage members at a higher resolution than previously reported[12], we crystallized an scFv of DH270.6 in complex with the synthetic Man$_9$-V3 glycopeptide and determined its structure at 2.17 Å resolution (Fig. 2a, b, Supplementary Fig. 1). We built the complete glycopeptide into the electron density map, along with the entire N332 glycan (Man$_9$GlcNAc$_2$) and the first three reducing-end carbohydrate residues at N301, the two $N$-acetylglucosamines (GlcNAc$_2$) and core mannose. As expected from its conformation in the Env trimer, Man$_9$-V3 formed a two-stranded β-hairpin with a bulge at the conserved $^{324}$GDIR$^{327}$ motif (Fig. 2b). The N332 Man$_9$ was well ordered, except for the terminal mannose residue of the D2 arm.

In the crystal structure, the antibody engaged the Man$_9$-V3 glycopeptide residues I322 to N332, the N332 glycan D1 and D3 arms, and the N301 GlcNAcs and core mannose. The conformations of the CDRs of the unliganded DH270.6[12] are the same as those seen in the complex; the superposition of unliganded and complexed DH270.6 gave a Cα root mean square deviation (r.m.s.d.) of 0.7 Å, showing that the antibody had not undergone large-scale conformational change upon binding to Man$_9$-V3. Differences in the side-chain conformations of R57$_H$, Y105$_H$, and Y106$_H$, which contact the peptide or N332 glycan, are discussed below.

The structure of the complex shows that there are two major regions of antibody–glycopeptide interactions: (1) the Env $^{324}$GDIR$^{327}$ motif (Fig. 1b), which is an important anchor point for binding N332-glycan dependent bnAbs, makes extensive contacts with the CDRH2 and CDRH3 loops of DH270.6, and (2) the N332-Man$_9$, which makes extensive contacts with the antibody CDRH3 and CDRL2 loops.

**Conformation of the Man$_9$-V3 glycopeptide**. Superposition of our structure on that of the BG505 SOSIP.664 in complex with PGT128[27] shows that the conformation of the synthetic glycopeptide is very close to that of the V3 loop in an intact, BG505 trimer (Fig. 3a), with a Cα r.m.s.d. of 1.31 Å. The conformation of the Man$_9$-V3 glycopeptide is similar to those of V3 loops in other HIV Env trimers from various strains, both when unliganded and

when bound with different V3-glycan-dependent antibodies (Fig. 3b, Supplementary Table 1), with Cα r.m.s.d.'s of less than 1.15 Å for all structures superposed. Glycopeptide–antibody contacts are therefore likely to represent those made with Env by the DH270 lineage antibodies. Moreover, the conformational similarity suggests that the synthetic V3 glycopeptide might be a useful immunogen, consistent with biochemical data previously reported[24].

Further evidence that Man$_9$-V3 is a good mimic of the V3 region of an intact trimer comes from a 3D reconstruction from EM images of negatively stained DH270.6 Fab in complex with the 92BR SOSIP.664 trimer used in our previous work on DH270.1[12] (Supplementary Fig. 2). As expected, three DH270.6 Fabs bound to each trimer, and there were no gross conformational differences between this trimer and the BG505 SOSIP.664 in complex with PGT128[27]. When we superposed the glycopeptide onto the V3 region of the trimer model, the DH270.6 scFv fit well into Env-proximal Fab density in the EM map (Supplementary Fig. 2).

We probed the Man$_9$-V3 structure in the absence of a binding partner by one-dimensional (1D) nuclear magnetic resonance (NMR) spectroscopy. The methyl region of the spectrum showed 11 well-resolved peaks with substantial chemical-shift dispersion (Supplementary Fig. 3A). Man$_9$-V3 has eight residues with methyl groups that could contribute to those peaks (5 Ile, 2 Thr, and 1 Ala) (Supplementary Fig. 3B). Although those residues would contribute a total of 13 methyl peaks, some probably overlap, reducing the distinguishable peaks to 11. Five isoleucine side chains contribute 10 of the 13 methyl groups in different regions of the glycopeptide (some are in the β-hairpin and others are in the GDIR bulge) (Supplementary Fig. 3B). The spectrum thus indicates distinct chemical environments for at least four of the five isoleucine residues (since the other three peaks from the spectrum could derive from the threonine or alanine side chains) and hence substantial non-random structure. The result is compatible with the interactions of isoleucine side chains seen in the crystal structure (e.g., Fig. 2c, Supplementary Fig. 3B). Addition of dithiothreitol to reduce the disulfide bond diminished the chemical-shift dispersion (Supplementary Fig. 3A), consistent with our inference from the structure that the disulfide, which is present in native Env, stabilizes the Man$_9$-V3 β-hairpin.

We generated additional evidence that Man$_9$-V3 has native-like ordered structure in solution by measuring the kinetics of DH270.6 Fab binding with the peptide and comparing the results with the kinetics of its binding with monomeric gp120 (Table 1, Supplementary Fig. 4A, 4B). The association rates for the two complexes were comparable, showing that Man$_9$-V3 did not need to undergo a slow folding step before stable association with the Fab; the equilibrium dissociation constants for peptide and gp120 were likewise similar. These data suggest that the conformation of the unbound peptide resembled that of the corresponding segment in intact gp120. The association rate constants are nonetheless lower than those of rapid, diffusion-limited protein associations, consistent with an initial, moderately slow, conformational adjustment in both cases.

**Interaction with V3 peptide and G57R mutation significance**. Somatic hypermutation is biased by the preference of the activation-induced cytidine deaminase (AID) for certain micro-sequence motifs ("hot spots")[28, 29]. Base substitution during somatic hypermutation also depends on surrounding sequences. Previous studies showed that the DH270 lineage acquired a G57R mutation in the CDRH2 loop between the DH270 unmutated common ancestor ("UCA") and the early intermediate, "IA4"[12]. Since the nucleotide sequence at the site of mutation (GGC) is an

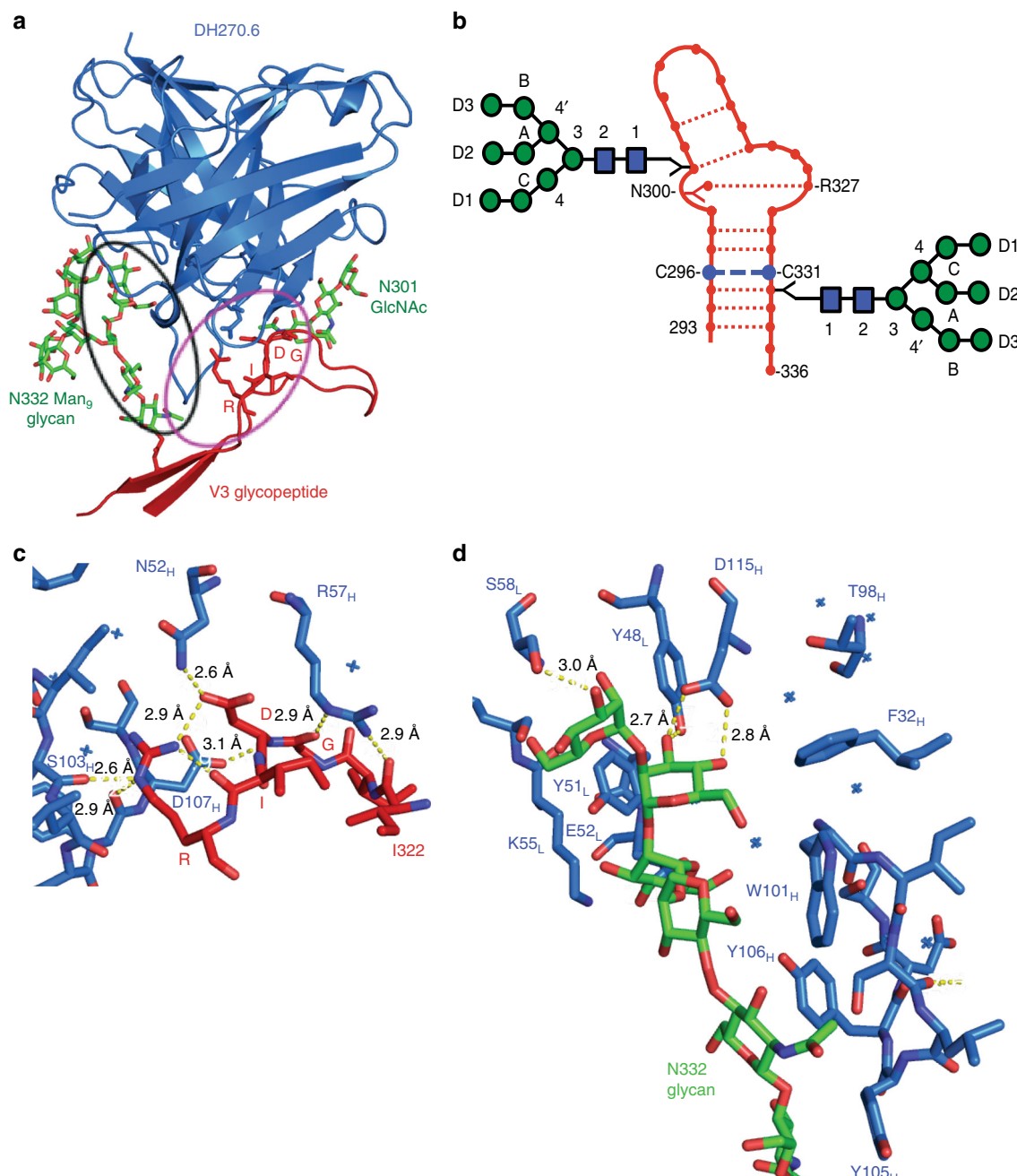

**Fig. 2** DH270.6-Env V3 loop structure. **a** DH270.6-Man$_9$-V3 crystal structure. The V3 region of the $^{324}$GDIR$^{327}$ motif is labeled, and glycans are shown as green sticks. The two major anchor points are circled. **b** Secondary structure of Man$_9$-V3. Each node (.) represents a peptide backbone carbonyl or amide group; dashed lines represent hydrogen bonds; Y represents a side chain; Man$_9$ glycans are shown as colored forks. A schematic diagram of the Man$_9$GlcNAc$_2$ glycan with the glycan moiety nomenclature is also shown. **c** The antibody-$^{324}$GDIR$^{327}$ motif hydrogen bonds. **d** Contacts between the antibody and Man$_9$ D1 arm

unlikely target for AID, this site is called an AID "cold spot" (a common cold spot microsequence generally represented by G̱RS, where R is a purine and S is either a C or a G and the underlined residue is the one that gets mutated), and a mutation at a site like this one is a rare event[28, 30]. Introducing the G57R mutation alone into the germline-reverted DH270 UCA conferred detectable binding to autologous Env and heterologous neutralization, implying that acquisition of the low probability mutation was critical to initiate the observed breadth of the lineage.

The side chain of R57$_H$ recognizes the 322–325 bulge in the V3 loop by donating hydrogen bonds to the main-chain carbonyls of I322 and G324. These contacts anchor the antibody–peptide interaction, much like the backbone contacts observed between the $^{324}$GDIR$^{327}$ motif and PGT128 CDRH2 and CDRH3 loops[9] or those between the side chain of D325 and the PGT124 CDRL1, CDRL3, and CDRH3 loops[11] (Fig. 2c, Supplementary Fig. 5A, 5B). Previous studies showed that an R57G$_H$ mutation in DH270. IA4 completely eliminates binding to HIV Env, illustrating the importance of this interaction[12]. The R57$_H$ interaction with the V3 loop differentiates $^{324}$GDIR$^{327}$-motif recognition by DH270 lineage members from its recognition by other well-characterized, N332-glycan dependent bnAbs[9, 11, 31].

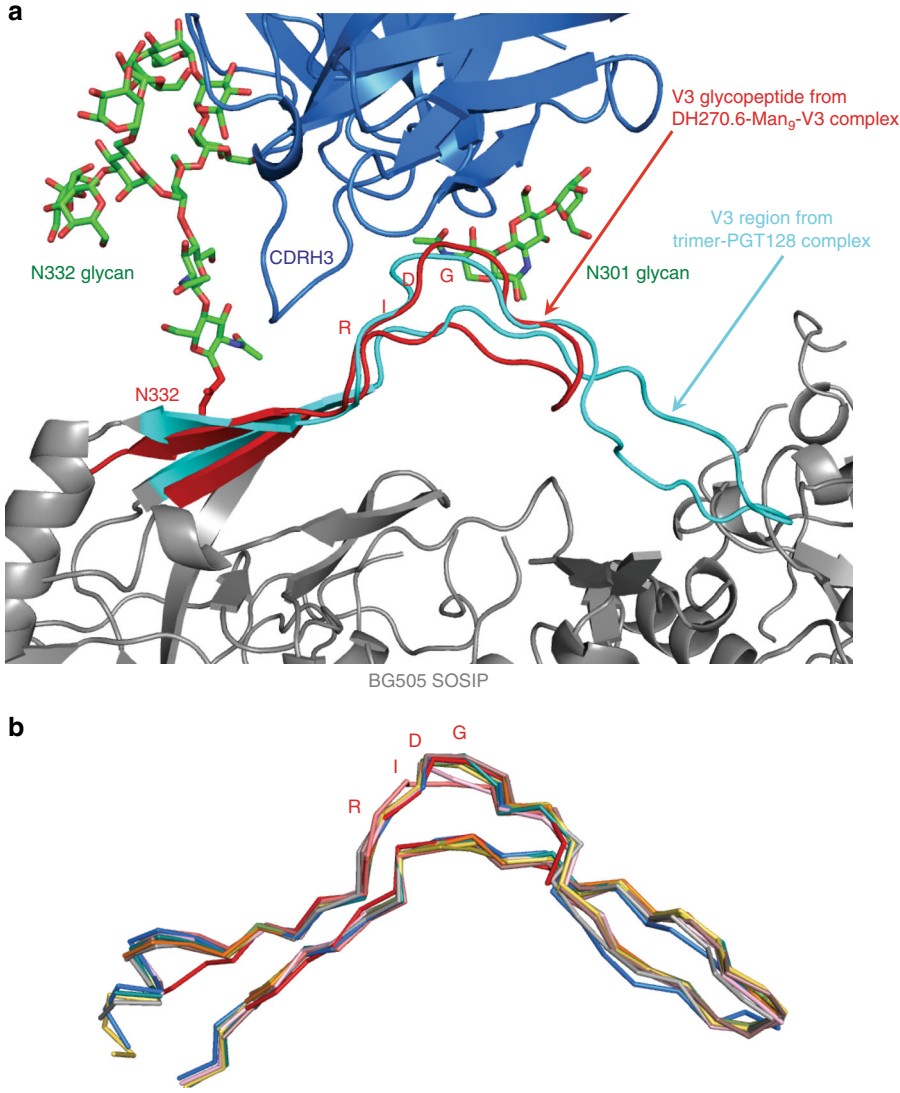

**Fig. 3** HIV trimer V3 region compared to V3 glycopeptide. **a** Superposition of the V3 glycopeptide (red) from the DH270.6-Man$_9$-V3 complex onto the V3 region (cyan) of the PGT128-BG505 complex (PDB ID: 5ACO). Glycans from the DH270.6-Man$_9$-V3 complex are shown as green sticks. **b** Superposition of V3 glycopeptide (red) from the DH270.6-Man$_9$-V3 complex onto the V3 region of other HIV Envs (both in complex, and unliganded) (PDB ID's 4ZMJ[47], 5CEZ[10], 5I8H[48], 5FYK[49], 5FYJ[49], 5T3S[50], 5V8L[51], 5T3Z[52])

**Table 1 Rate constants and equilibrium dissociation constants for binding of DH270.6 Fab with Man$_9$-V3 and with gp120**

|  | $k_a$ (M$^{-1}$s$^{-1}$) | $k_d$ (s$^{-1}$) | $K_D$ (μM) |
|---|---|---|---|
| Man$_9$-V3 + DH270.6 Fab | $1.0 \pm 0.05 \times 10^4$ | $(0.07 \pm 0.005)$ | $7 \pm 0.4$ |
| gp120 + DH270.6 Fab | $0.9 \pm 0.2 \times 10^4$ | $(0.01 \pm 0.04)$ | $1.2 \pm 0.3$ |

Association rate constants ($k_a$) and equilibrium dissociation constants ($K_D$) were determined from duplicate BLI measurements at five concentrations (Supplementary Fig. 4). Dissociation rate constants were calculated as the products of $k_a$ and $K_D$. Errors are SEM

In addition to interactions through R57$_H$, DH270.6 forms an extended network of polar contacts with the $^{324}$GDIR$^{327}$ motif, further distinguishing antibodies in this lineage from other V3-glycan bnAbs. In particular, N52 in the CDRH2 loop donates a hydrogen bond to the D325 carboxylate, fixed in its side-chain conformation by a salt bridge with R327, and D107 in the

CDRH3 loop accepts a hydrogen bond from the D325 backbone amide (Fig. 2c). The R327 guanidinium group also hydrogen bonds with S103$_H$. Mutating D325 to Arg leads to a complete loss of detectable binding to HIV-1 Env, confirming the importance of this residue for antibody recognition[12]. The Env R327A mutant has only a slightly lower affinity for DH270.6 than its wild-type counterpart (Fig. 4a), while the S103G mutation weakens the Env interaction twofold (Fig. 4b, Table 2). Thus, for the DH270 lineage, contacts with the GDIR peptide backbone (by R57$_H$ and D107$_H$) and with the D325 side chain (by N52$_H$) are essential sites of interaction with the conserved anchor region of Env V3; the antibody–R327 interaction is a less critical, secondary contact point.

**Interaction of DH270.6 with N332 Man$_9$ glycan.** N332-glycan-dependent bnAbs typically have a long CDR loop that can contact both glycan and protein residues. DH270 lineage members have shorter CDRs than other bnAbs of the same class[12], but it is nonetheless their longest CDR, the CDRH3, that contacts the V3 peptide. In our Man$_9$-V3:DH270.6 structure, the N332 glycan and

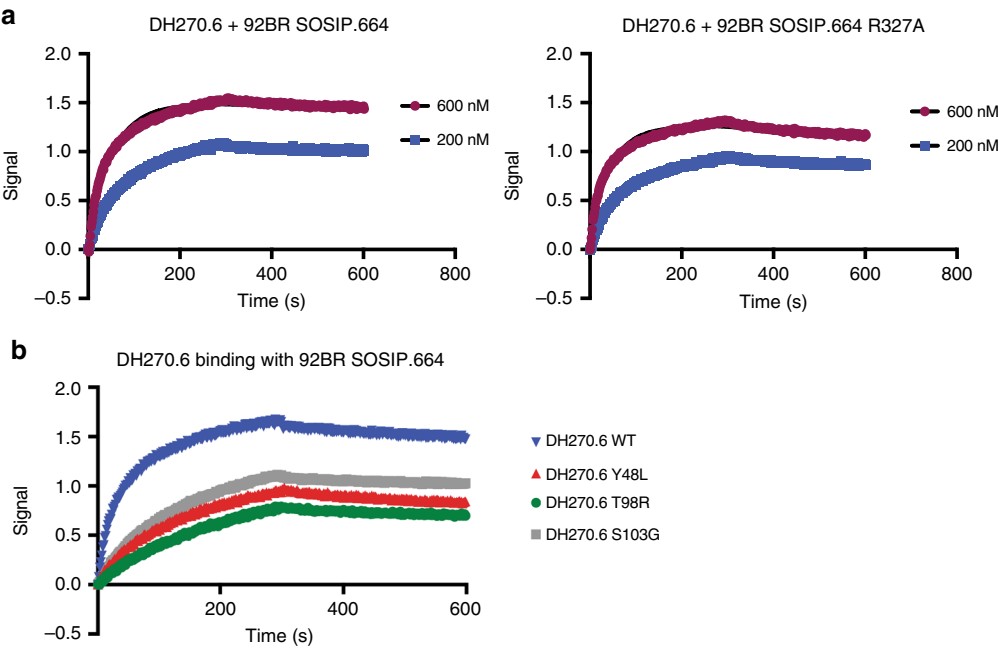

**Fig. 4** Binding of DH270.6 and Env. Biolayer interferometry association and dissociation curves are shown for **a** wild-type ($K_D = 2.76$ nM) and R327A mutant ($K_D = 7.64$ nM) forms of 92BR SOSIP.664 Env tested with DH270.6 Fab at the indicated concentrations and **b** wild-type and mutant forms of DH270.6 Fab binding to wild-type 92BR SOSIP.664

| **Table 2 Dissociation constants of Fab complexes with wild-type 92BR SOSIP.664** | |
|---|---|
| | $K_D$ (nM) |
| DH270.6 WT | 11 ± 1 |
| DH270.6 Y48L$_L$ | 44 ± 2 |
| DH270.6 T98R$_H$ | 46 ± 2 |
| DH270.6 S103G$_H$ | 19 ± 1 |
| Errors are SEM, determined from duplicate measurements | |

the V3 peptide sandwich the CDRH3 loop, which has extensive interactions, both polar and non-polar, with the glycan (Fig. 2d).

One key polar interaction appears to be a bidentate contact between D115$_H$ in CDRH3 and the C-mannose moiety of the D1 arm of N332-Man$_9$. Similar contacts with D1-arm mannoses have been seen in PGT124 and PGT128 complexes[9, 11]. Mutation of D115$_H$ to alanine in DH270.1 reduces Env affinity[12]. Spatially adjacent to the D115$_H$-mannose interaction is a contact from Y48$_L$ of the CDRL2 loop, which hydrogen bonds with the same mannose. We mutated Y48$_L$ to leucine, which is found in the UCA of this lineage, and found fourfold lower affinity for 92BR SOSIP.664 Env trimer (Fig. 4b, Table 2). The N332 glycan also stacks against Y51$_L$, a germline-encoded residue, which is nearby. Because the Y51$_L$ residue is germline encoded (i.e. present in the DH270 UCA) and is in the same conformation as in the later antibody lineage members, it would not have been a principal contributor to affinity maturation.

DH270.6 residues engaged in hydrophobic contacts between CDRH3 and the N332 Man$_9$ glycan include W101$_H$, Y105$_H$, and Y106$_H$ (Fig. 2d). The Y105$_H$ and Y106$_H$ side chains are disordered in the unliganded DH270.6 structure[12]. In the complex however, Y105$_H$ inserts into a groove near H330 of V3 and the core GlcNAc of the N332 glycan while Y106$_H$ is in hydrophobic contact with the second GlcNAc and the adjoining

mannose residue. Mutation of these antibody residues reduces Env binding[12]. E52$_L$ and S58$_L$ also contact D1 arm mannose residues, but these, like Y51$_L$, are germline-encoded, and probably do not provide an advantage to later antibody lineage members for Env recognition.

DH270 lineage antibodies also interact with the D3 arm of the N332 Man$_9$ glycan and the GlcNAcs and core mannose of the N301 glycan; the latter interaction is less extensive, consistent with observations showing it to be dispensable for Env binding and viral neutralization[12]. The side chain of D31$_H$ in the antibody CDRH1 loop appears to contact the terminal mannose of the D3 arm, although density for this arm is not as strong as for the D1 arm mannoses. We previously showed that mutating this residue in the antibody reduces Env affinity[12]. Interaction with this terminal, D3-arm mannose, along with those of the D1 arm, shows why high-mannose glycans are critical for interactions with late members of this antibody lineage.

**DH270.3-Man$_9$ complex identifies a critical mutation.** Antibody DH270.3, in a different branch of the lineage than DH270.6, is both less potent against autologous and heterologous viruses and has a narrower neutralization spectrum (Fig. 1a). We determined a crystal structure of DH270.3 bound with a synthetic, biotinylated Man$_9$ glycan[24]. In the density map at 2.85 Å resolution, all of the Man$_9$ sugar moieties were well resolved, and the refined model showed that they superpose well on the N332-Man$_9$ glycan from our structure of DH270.6 bound with Man$_9$-V3, except for the terminal mannose of the D2 arm, which does not contact either antibody (Fig. 5a). The conformation of the N332-Man$_9$ glycan in our structures does, however, differ from that of other N332-glycan bnAb complexes (Supplementary Fig. 6), illustrating the conformational flexibility of these carbohydrates and showing that antibody CDRs can induce conformational shifts in HIV Env glycans. In contrast, the conformation of the V3 region, which is probably more rigid than the glycans, varies little from structure to structure. In any case, the DH270.3-Man$_9$ structure confirms the contacts we

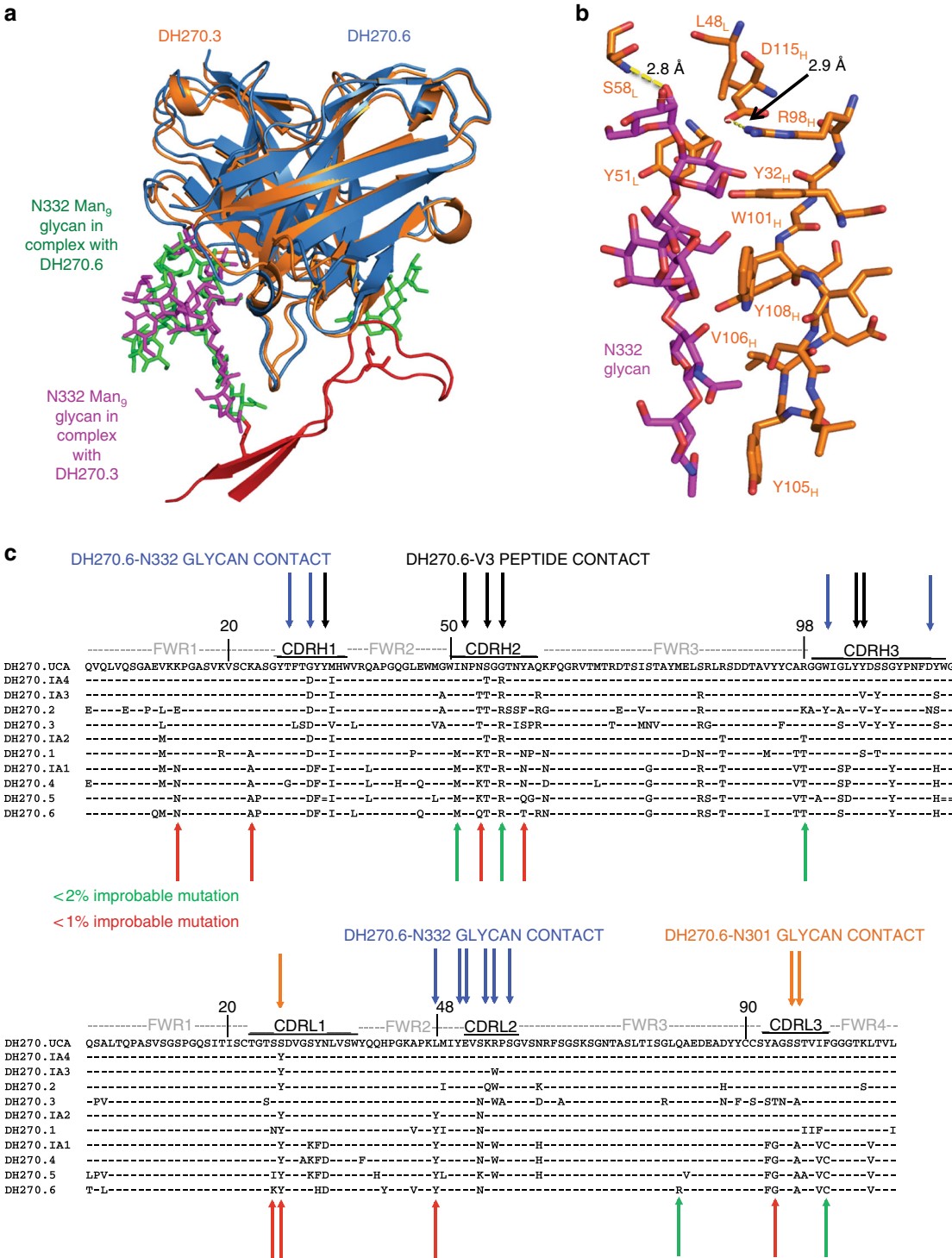

**Fig. 5** DH270.3-Man$_9$ crystal structure. **a** The structure of DH270.3 (orange) in complex with Man$_9$ (magenta), superposed on DH270.6 (blue) in complex with Man$_9$-V3 (red), with glycans shown as green sticks. **b** Contacts made by DH270.3 and Man$_9$. **c** Sequence alignments of DH270 lineage members with glycan and peptide contact sites indicated. CDR loops, framework regions, and improbable mutations are also shown

observed between DH270.6 and the N332-glycan, and shows a conserved mode of glycan recognition by these clonally related antibodies.

The one notable difference between the two structures is at D115$_H$. This residue of the DH270.3 antibody forms a salt bridge with R98$_H$, rather than the bidentate interaction with a terminal mannose seen in the DH270.6 complex. (Figs. 2d, 5b, Supplementary Fig. 7). DH270.6 has threonine instead of arginine at

position 98$_H$. When we reverted T98$_H$ in DH270.6 to Arg, we detected only negligible residual binding with 92Br SOSIP.664 Env trimer (Fig. 4b). Thus, the arginine appears to "distract" D115$_H$ from interacting with the glycan, and its somatic mutation to Thr (another low probability event: Fig. 5c) correlates with development of breadth, probably by allowing D115$_H$ to interact with a terminal mannose instead of with another residue in the antibody.

## Discussion

The structure of the DH270.6-Man$_9$-V3 complex shows that this glycopeptide accurately mimics the V3 region of an intact native-like HIV Env trimer—a result consistent with previous data[24]. The high specificity of these antibodies for the peptide motifs and the homogeneity of synthesized glycopeptides makes them useful tools for dissecting protein—protein interactions and for immunogen design. The Man$_9$-V3 glycopeptide used in our studies included a larger portion of the V3 region than present in previously reported V3 mimics[3–7], and also included two critical glycans, at N332 and N301, often contacted by V3-glycan "supersite" bnAbs. Inclusion of glycans in a peptide mimetic is clearly important for analyzing protein contacts in which they are involved. Indeed, the Man$_9$-V3 glycopeptide in our crystal structures was also used to isolate the DH270.6 bnAb by labeling the BCR of memory B cells from the HIV-1 infected donor[12].

The goal of an HIV-1 vaccine is to elicit neutralizing Abs of significant breadth. Eliciting an antibody like DH270.6 would be more favorable than eliciting one like DH270.3, with a narrower spectrum of neutralizing activity. Our structures showed that the R98T$_H$ mutation, which is found in many N332-glycan dependent bnAbs (Supplementary Fig. 8) and occurs at a disfavored AID site, was a critical step in the affinity maturation pathway. The mutation allowed the conserved residue D115 in the antibody heavy chain to interact favorably with the C-mannose moiety in the D1 arm of N332-Man$_9$, while R98$_H$ and L48$_L$, found in the UCA, DH270.3, and several early intermediates, anchor D115$_H$ in a rotamer that does not allow the glycan contact. DH270.2, which is close to DH270.3 in the lineage, has lysine at position 98$_H$, which could in principle form a salt bridge with D115$_H$, like R98$_H$ in the other antibodies. Analogous contacts with the D1 arm mannose moiety are present in Env complexes of PGT128 (by D95a$_L$) and PGT124 (by R100$_H$) (Supplementary Fig. 5C, 5D)[9, 11]. Loss of neutralization accompanies mutation of these residues. PGDM21, which has a CDRH3 of the same length as those of DH270 lineage members, has been reported to bind gp120 with both high mannose and complex glycans at N332; its binding also depends on the D1 arm mannose moiety[13]. The R98$_H$T mutation could thus have been an important event in the development of breadth in the DH270 lineage.

Another previously identified, improbable mutation, G57R$_H$, which occurred between the UCA and IA4 of the lineage and has generally not been found in other N332-glycan bnAbs identified to date[9, 11, 13, 14, 32] (Supplementary Fig. 8), appears to have been a critical initial step in expansion of the DH270 bnAb lineage[12]. The polar hydrogen bonds between R57$_H$ and backbone carbonyls in the conserved 322–325 bulge show that an antibody with shorter CDR loops than those of PGT124 and PGT128 can still penetrate the glycan shield to make conformation specific contacts with the Env protein. Other N332-glycan-dependent bnAbs with short CDR loops have been identified, e.g. the PCDN antibody lineage[32]. These antibodies have CDRH3 loops that are only two residues longer than those of the DH270 lineage and have shorter CDR loops than PGT124 and PGT128 (Supplementary Fig. 8). Moreover, PCDN antibodies have been shown to have low levels of autoreactivity. Thus, the correlation of longer CDR loops with auto- and polyreactivity[33–35] may not be a major barrier for eliciting V3-glycan-directed antibodies.

Like most bnAb lineages, DH270 developed breadth only several years after infection, at least in part because of the two improbable somatic mutations needed for specific anchoring at two complementary positions on Env. The G57R$_H$ mutation, early in the lineage, allowed IA4 (or an analogous precursor) to latch onto the V3 loop backbone; the subsequent R98T$_H$ mutation in the DH270.6 branch allowed binding to a high mannose moiety of the N332 glycan. Persistent encounters with antigen during several years of chronic infection were presumably critical in selecting for this sequence of low probability mutational events. Repeated immunization may likewise be essential for bnAb induction.

Immunizations with Man$_9$-V3 alone have not yielded bnAb responses[24]. Man$_9$-V3 may, however, be useful in a prime-boost format, in which Man$_9$-V3 is used to prime precursors of V3-glycan B-cell lineages, followed by boosts with multimerized Man$_9$-V3 to augment T-cell responses or with sequential Envs found in HIV-1-infected individuals who have developed V3-glycan bnAbs. Alternatively, it might be useful to use Man$_9$-V3 in conjunction with a monomer containing a T-cell helper epitope, such as a glycoconjugate of the non-natural pan DR epitope[36], to improve carbohydrate-specific antibody responses by immunization with a glycopeptide.

The Man$_9$-V3 glycopeptide used in these studies maintains a substantial amount of structure in solution even when not bound to a specific antibody, and when bound it accurately mimics the native conformation, with high affinity for a potentially desired bnAb such as DH270.6. The structure we determined suggests ways to design a modified Man$_9$-V3 with increased stability. For example, H330 could be replaced with a residue, such as glutamine, that could hydrogen bond with Thr297, to increase stability of the hairpin (Supplementary Fig. 3C). The absence of distracting epitopes in Man$_9$-V3 suggests that minimal antigens of this kind might be suitable candidate immunogens in regimes involving long-term antigenic exposure.

## Methods

**Expression and purification of DH270 lineage members.** The heavy- and light-chain variable and constant domains of the DH270.3 and DH270.6 Fabs were cloned into a modified pVRC-8400 expression vector using Not1 and Nhe1 restriction sites and the tissue plasminogen activator signal sequence. The DH270.6 scFv was cloned into the same vector as above. The C terminus of the heavy-chain constructs and scFv contained a noncleavable 6× histidine tag. Site-directed mutagenesis was performed using manufacturer's protocols (Stratagene) to introduce mutations into the CDR regions of DH270.3 and DH270.6 Fabs. Fabs and the DH270.6 scFv were expressed and purified as described previously[12, 37]. Briefly, Fabs and the DH270.6 scFv were expressed using transient transfection of HEK 293T cells (ATCC: CRL-3216) using linear polyethylenimine (PEI) following the manufacturer's suggested protocol. After 5 days of expression, supernatants were clarified by centrifugation. His-tagged Fabs were loaded onto Ni-NTA superflow resin (Qiagen) preequilibrated with Buffer A (10 mM Tris, pH 7.5, 100 mM NaCl), washed with Buffer A + 10 mM imidazole, and eluted with Buffer A + 350 mM imidazole. Fabs were then purified by gel filtration chromatography in Buffer A using a superdex 200 analytical column (GE Healthcare).

**Synthesis of Man$_9$-biotin, Man$_9$-V3-biotin and Man$_9$-V3.** Man$_9$-biotin and Man$_9$-V3-biotin were synthesized as described earlier[24]. Man$_9$-V3 was synthesized following an identical strategy as previously described. A full description of the chemical synthesis is detailed in the Supplementary Methods. Briefly, using our one-flask aspartylation/deprotection protocol, Man$_9$GlcNAc$_2$ glycosyl amine **1** was joined to the free carboxylic acid side chain at position 301 on fragment **2** and at position 332 on fragment **3**, followed by trifluoroacetic acid treatment to provide glycopeptide thioester **4** and N-terminal cysteinyl glycopeptide **5**. These two fragments were then joined by native chemical ligation immediately followed by cyclization via disulfide formation to afford Man$_9$-V3 (**6**).

**Crystallization, structure determination, and refinement.** The His-tagged DH270.3 Fab was mixed with 6 molar excess of biotinylated-Man$_9$ glycan at a final complex concentration of 25 mg/ml and the DH270.6 scFv was mixed with 3 molar excess of Man$_9$-V3 at a final complex concentration of 20 mg/ml for crystallization trials. Crystals were grown in 96-well format using hanging drop vapor diffusion and appeared after 24–48 h at 20 °C. Crystals were obtained in the following conditions: 30% PEG 8000, 10 mM CHES, pH 9.0 and 1 M NaCl for the DH270.6-Man$_9$-V3 complex; and 1 M Na/K phosphate, pH 5.6 for the DH270.3-Man$_9$ complex. All crystals were harvested and cryoprotected by the addition of 20–25% glycerol to the reservoir solution and then flash-cooled in liquid nitrogen.

Diffraction data were obtained at 100 K from beam line 24-ID-E at the Advanced Photon Source using a single wavelength. Data sets from individual crystals were processed with XDS[38]. Molecular replacement calculations for the free scFv and Fab were carried out with PHASER[39], using published DH270.6 and DH270.3, respectively, (Protein Data Bank (PDB) ID 5TQA and 5TPL) as the

## Table 3 Crystallographic statistics

| | DH270.6-Man$_9$-V3 | DH270.3-Man$_9$ |
|---|---|---|
| Data collection | | |
| Resolution (Å) | 60.14–2.17 (2.25–2.17) | 64.04–2.85 (2.95–2.85) |
| Space group | P 4$_1$ 2 2 | I 4$_1$ |
| Unit cell a,b,c (Å) | 68.0, 68.0, 128.8 | 128.1, 128.1, 91.9 |
| Unit cell $\alpha$, $\beta$, $\gamma$ (°) | 90, 90, 90 | 90, 90, 90 |
| Total reflections | 77,943 | 28,007 |
| Unique reflections | 15,878 | 17,375 |
| Redundancy | 4.9 (4.3) | 3.8 (3.8) |
| Completeness (%) | 95.1 (96.2) | 99.6 (99.7) |
| $<I/\sigma_I>$ | 7.65 (1.12) | 11.66 (1.35) |
| $R_{merge}$ | 22.7 (149.7) | 13.0 (105.7) |
| Refinement | | |
| $R_{work}/R_{free}$ (%) | 22.1/27.8 (31.9/37.7) | 20.4/23.1 (29.4/33.4) |
| No. atoms | | |
| Protein | 2019 | 3256 |
| Ligand | 155 | 132 |
| Water | 96 | 8 |
| R.M.S. deviations | | |
| Bond lengths (Å) | 0.005 | 0.006 |
| Bond angles (°) | 1.04 | 1.02 |
| B-factors (Å$^2$) | | |
| Protein | 38.00 | 62.30 |
| Ligand | 30.00 | 43.40 |
| Solvent | 41.00 | 39.50 |

Statistics for the highest-resolution shell are shown in parentheses

starting models. The Fab model was separated into its variable and constant domains for molecular replacement.

Refinement was carried out with PHENIX[40], and all model modifications were carried out with Coot[41]. During refinement, maps were generated from combinations of positional, group B-factor, and TLS (translation/libration/screw) refinement algorithms. Secondary-structure restraints were included at all stages. Structure validations were performed periodically during refinement using the MolProbity server[42]. The final refinement statistics are summarized in Table 3.

**Expression and purification of Envs.** The SOSIP.664 constructs were cloned into the same pVRC-8400 vector described above for Fabs and any mutations were added by site-directed mutagenesis using manufacturer's protocols (Stratagene). SOSIP.664 constructs were transfected along with a plasmid encoding the cellular protease furin at a 3:1 Env:furin ratio in 293F cells, adapted to grow in suspension from HEK293T cells (ATCC: CRL-3216). The expression construct of furin was kindly provided by Bing Chen (Boston Children's Hospital, Boston, MA). The cells were allowed to express soluble SOSIP trimers for 5–7 days, and gp120 monomers were expressed for 5 days. Culture supernatants were collected and cells were removed by centrifugation at 3800 × g for 20 min, and filtered with a 0.2 μm pore size filter. Envs were purified by flowing the supernatant over a lectin (*Galanthus nivalis*) affinity chromatography column overnight at 4 °C. The lectin column was washed with 1× PBS, followed by a second wash with 1× PBS and 0.5 M NaCl and proteins were eluted with 1× PBS supplemented with 0.5 M methyl-α-D-manno-pyranoside and 0.5 M NaCl. The eluate was concentrated and loaded onto a Superdex 200 10/300 GL column (GE Life Sciences) pre-equilibrated in a buffer of 2.5 mM Tris, pH 7.5, 350 mM NaCl and 0.02% sodium azide for BLI and in 10 mM Hepes, pH 8.0, 150 mM NaCl and 0.02% sodium azide for negative stain EM, to separate the trimer-size oligomers from aggregates and gp140 monomers and to separate gp120 monomers from aggregates.

**Electron microscopy.** Purified 92BR SOSIP.664 trimer was incubated with a 6 molar excess of DH270.6 Fab at 4 °C for 1.5 h. A 2.5 μl aliquot containing ~0.01 mg/ml of the Fab-92BR SOSIP.664 complex was applied for 30 s onto a carbon-coated 400 Cu mesh grid that had been glow discharged at 5 mA for 2 min, followed by negative staining with 0.7% uranyl formate for 20 s. Samples were imaged using a FEI Tecnai T12 microscope operated at 120 kV and a magnification of ×52,000, yielding a pixel size of 2.13 Å at the specimen plane. Images were acquired with a Gatan 2 K CCD camera using a nominal defocus of 1500 nm at 10° tilt increments, up to 50°. The tilts provided additional particle orientations to improve the image reconstructions.

**Negative stain image processing and 3D reconstruction.** Particles were picked semi-automatically using EMAN2 and put into a particle stack. Initial, reference-free, two-dimensional (2D) class averages were calculated, and particles corresponding to complexes (with three Fabs bound) were selected into a substack for determination of an initial model. The initial model was calculated in EMAN2 using threefold symmetry and EMAN2 was used for subsequent refinement using threefold symmetry. In total, 12,941 particles were included in the final reconstruction for the 3D average of 92BR SOSIP.664 trimer complex with DH270.6. The resolution of the final model was estimated using a Fourier Shell Correlation (FSC) cut-off of 0.5.

**Model fitting into the EM reconstructions.** The cryo-EM structure of PGT128-liganded BG505 SOSIP.664 (PDB ID: 5ACO) and crystal structure of DH270.6 were placed by inspection into the EM density and refined by using the UCSF Chimera 'Fit in map' function.

**Sample preparation for the NMR experiments.** Man$_9$-V3 was dissolved in 50 mM phosphate (Pi) buffer (pH 7.5) to a final concentration of ~1 mM in 360 μl. Ten percent (v/v) D$_2$O was added for the NMR lock. DTT was added to a final concentration of 20 mM to reduce the disulfide bond.

**NMR data acquisition, processing and analysis.** The NMR experiments were performed at 14.1 T on a Bruker Avance III HD spectrometer operating at 600.13 MHz $^1$H, 150.90 MHz $^{13}$C, and 60.81 MHz $^{15}$N frequencies, equipped with a cryogenically cooled probe head. All data were collected at 298.0 K. The NMR 1D data sets were processed and analyzed using *nmrPipe* and *nmrDraw*, respectively[43].

**Biolayer interferometry.** Kinetic measurements of Fab binding to the SOSIP.664 constructs were carried out using the Octet QK$^e$ system (ForteBio); 0.2 mg/ml of each His-tagged Fab was immobilized onto a anti-Human Fab-CH1 biosensor until it reached saturation. The SOSIP.664 trimers were tested at concentrations of 0.1 μM to 0.6 μM. A reference sample of buffer alone was used to account for any signal drift that was observed during the experiment. Association and dissociation were each monitored for 5 min. All experiments were conducted in the Octet instrument with agitation at 1000 rpm.

Measurements of the kinetics of DH270.6 Fab binding to Man$_9$-V3 and gp120 were carried out using a BLItz instrument (ForteBio). To measure DH270.6 binding to Man$_9$-V3, 0.2 mg/ml of biotinylated Man$_9$-V3 was immobilized on a streptavidin biosensor at saturation. To measure gp120 binding to the DH270.6 antibody, 0.2 mg/ml of DH270.6 Fab was immobilized on an anti-Human Fab-CH1 biosensor at saturation. The binding of Man$_9$-V3 and gp120 were tested at Fab concentrations of 40, 15, 5, 1.5, and 0.25 μM, and gp120 concentrations of 25, 15, 5, 1.5, and 0.25 μM, respectively, at 22 °C. Association and dissociation were each monitored for 5 min.

Analyses were performed using nonlinear regression curve fitting using the Graphpad Prism software, version 7.

**Antibody sequence analysis.** Probabilities of amino acid mutations prior to selection were calculated using the ARMADiLLO program[44]. Briefly, ARMA-DiLLO simulates the somatic hypermutation process and uses these simulations to estimate the probability of an amino acid substitution occurring, in the absence of selection, from the unmutated common ancestor given the number of mutations observed in the antibody of interest. Reconstruction of the DH270 clonal genealogy and corresponding clonal lineage tree was generated with the software package Cloanalyst[45, 46].

**Protein structure analysis and graphical representations.** The complexes analyzed in this study were superposed by least squares fitting in Coot or PyMol. All graphical representations with protein crystal structures, including stereo images, were made using PyMol.

**Data availability.** Coordinates and structure factors for Fab DH270.3, bound with Man$_9$ glycan, and DH270.6 scFv, bound with Man$_9$-V3, have been deposited in the Protein Data Bank with accession code 6CBJ, and 6CBP, respectively. Sequences for the DH270.3 and DH270.6 immunoglobulin chains have been deposited previously (Genbank accession codes KY354944, KY354957, KY354948, and KY354961). Probabilities of amino acid mutations were calculated using the ARMADiLLO program (Preprint at https://www.biorxiv.org/content/early/2018/02/09/262592, 02092018, https://doi.org/10.1101/262592). All other data supporting the findings of this study are available within the article and its Supplementary Information files, or are available from the authors upon request.

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

## Acknowledgements

We thank James Chou for advice on the NMR experiments and their interpretation. We thank staff at Northeastern Collaborative Access Team (NE-CAT) x-ray beam line 24 ID-C (Advanced Photon Source). NE-CAT is funded by NIH grant P41 GM103403; the Pilatus 6M detector on 24-ID-C is funded by an NIH-ORIP HEI grant (S10 RR029205). This research used resources of the Advanced Photon Source, a U.S. Department of Energy (DOE) Office of Science User Facility operated for the DOE Office of Science by Argonne National Laboratory under Contract No. DE-AC02-06CH11357. The research was supported by National Institute of Allergy and Infectious Diseases Grant AI100645 (Center for HIV/AIDS Vaccine Immunology-Immunogen Discovery). D.F. acknowledges NIH fellowship 1F32-AI-116355 and Mathilde Krim Fellowship in Basic Biomedical Research (109502-61-RKVA) from amfAR. R.R.M. acknowledges support from

Medical Scientist Training Program (MSTP) training grant T32GM007171 (R.R.M.) and Ruth L. Kirschstein National Research Service Award F30-AI122982. A.P. acknowledges support from NIH grant AI127193. S.C.H. is an investigator in the Howard Hughes Medical Institute.

## Author contributions

D.F. and S.C.H. coordinated and designed the study, analyzed and evaluated the data, and wrote and edited the manuscript and figures; D.F., M.S.L., A.P., T.T., and J.O.Z. performed experiments, analyzed the data, and produced the figures; M.S.L. performed electron microscopy studies; A.P. performed NMR experiments, T.T. performed BLI experiments, J.O.Z. expressed proteins and performed sequence analyses, M.B. directed the memory B-cell cultures and functional studies of monoclonal antibodies; R.R.M. isolated the antibodies through antigen-specific single-cell sorting; B.A., W.E.W., and S. D. synthesized the $Man_9$-V3 glycopeptide; K.W. performed computational analyses; M.S. L., K.W., A.P., R.R.M., and M.B. edited the manuscript; R.R.M., M.B., and B.F.H. provided intellectual contributions.

## Additional information

**Competing interests:** The authors declare no competing interests.

