## [Peer Review File(PDF 469 kb) · Nature Communications]

Reviewers' comments:

Reviewer #1 (Remarks to the Author):

To the authors

This paper by Fera et al/the Harrison group describes the structures of broadly neutralizing and less broadly neutralizing DH270 lineage Nabs with a V3 glycopeptide, revealing details of how these MAbs engage principally with the GDIR motif and the adjacent N332 high mannose glycan D1 arm along with other local glycan and peptide structures. An analysis of the lineage shows how early in the lineage a G57R mutation allows contact with the GDIR motif at the V3 base and

that the broader lineage member emerges by eliminating D115 (CDRH3 c-terminus) self-contact with Y98 at the end of heavy chain framework 3 by R98T mutation. Both of these key mutations are rare AID mutations and may therefore represent a barrier to induction of these bnAbs. The authors conclude by suggesting that V3 glycan constructs used for the co-crystals might be useful immunogens.

Overall, the paper is succinct, clear and well written. A few comments and largely minor criticisms are as follows:

*Fig 5C is a nice summary of the findings, but it would be better if framework and CDRs were included for orientation purposes

*A paper by Sok-Burton in Immunity 2016 PGDM12, 14, 21 targeting the high mannose patch GDIR region could probably be cited. Are their features similar to DH270 lineage? Similarly, there are bnAbs from the VRC reported in JVirolgy Longo et al. Dec 2016. There may be others and they should be cited as a way to mine similarities and differences as may be relevant for vaccine design. Possibly these could be aligned as in Fig 5C to show similarities and differences to assist in the narrative in the discussion (page 12 line 251) and help make the point regarding differing CDR lengths mentioned. Even if this information is present in a preceding paper, it is arguably worth some extra space here to emphasize the point.

*I could not fully understand Fig. 3. Some labeling of residues might help to orient the reader. The same goes for Supp figures 3 and 4

*I was not clear on the meaning of the NMR data. Can the interpretations be explained a bit more explicitly?

*I also felt that a sentence or two could be added to explain why the key mutations (above) are cold spots. Why is this so? References are given, but arguably the story is incomplete without at least some brief explanation as to why cytidine deaminase makes some mutations more difficult to achieve than others.

*line 247 "Tyr48L is actually Leu48 in the UCA and DH270.3 and is not shown in fig 5B or mentioned in results and maybe it should be.?"

*a mutation of R98T in DH270.3 might be a nice complement to the KO mutation done in DH270.6 to make the point more concrete.

*It's not immediately clear how the V3 glycan construct could be used as an immunogen and in fact it was used in ref 17 and did not induce any Nab responses. It does seem reasonable to suggest that it could be useful in a prime boost format. One does wonder if sufficient helper epitopes would be present to make it an effective immunogen without e.g. PADRE.

Reviewer #2 (Remarks to the Author):

The authors describe the structure of a broadly neutralizing HIV-1 antibody, DH270.6, in complex with its glycopeptide epitope as well as the structure of a less broad intermediate clonal family member in complex with a glycan. The structures reveal various details on evolution of the antibody lineage and confirm that the epitope of the antibody can be recapitulated by a synthetic glycopeptide. Both findings are of interest to the field. The authors previously published a structure of the a member of the DH270 lineage as well as the lineage itself, but the current structures are at higher resolution and the presence of the glycopeptide allows for a more detailed structural interpretation of the affinity maturation pathways.

Specific comments:

1. Discussion. Lines 230-231. The authors imply that the glycopeptide in solution assumes the same structure of as this region in the Env protein. However, the glycopeptide might very well attain this particular structure using the antibody as the template (induced fit). Considering the central role of the glycopeptide in this manuscript it is important to flesh out this subject a bit more. The NMR data are OK, but not very convincing to imply that the peptide is conformationally stable in solution. Additional in solution assays such as circular dichroism or other could provide more information on the structure of the glycan peptide in solution. Isothermal titration calorimetry could be used to assess whether antibody binding is accompanied by substantial conformational changes, and/or increase in order.
2. Furthermore, the impact of the manuscript could be enhanced substantially by using the structure to design more conformationally stable versions of the glycopeptide.
3. Results. In order to make more general statements on V3 broadly neutralizing antibody recognition, it would be good to include figures comparing the structures of DH270.3, DH270.6 and other antibodies targeting this region such as PGT124 and PGT128. Does the 332 glycan assume the same structure in all complexes? And the 301 glycan? It would be good to show this visually. Similarly for the peptide component (i.e. GDIR). It would probably be appropriate to add such figure to Fig. 3. The authors could generate an overlay of the individual glycans and GDIR for DH270.3, DH270.6, PGT124, PGT128 and possibly

others and compare the contact residues in epitope and paratope.

4. Introduction. About half of the introduction is a summary of results and conclusions, not introduction. The actual introduction is extremely short.

5. Introduction. Lines 59-60. Most glycans are indeed heterogeneous, but it appears from studies from the Crispin and Paulson labs, that the HIV Env glycans are more homogeneous than those on other glycoproteins. Glycans at many positions on Env are found exclusively as oligomannose, or even only Man9 glycans.

Fig. 2B. The size of the glycans in the left part of the figure is not proportional to the size of the peptide. I would suggest to size glycans and peptide proportionally and remove the right part of the figure (but transfer the labels to the glycans in the left side of the figure).

Reviewers' comments:

Reviewer #1 (Remarks to the Author):

To the authors

This paper by Fera et al/the Harrison group describes the structures of broadly neutralizing and less broadly neutralizing DH270 lineage Nabs with a V3 glycopeptide, revealing details of how these MAbs engage principally with the GDIR motif and the adjacent N332 high mannose glycan D1 arm along with other local glycan and peptide structures. An analysis of the lineage shows how early in the lineage a G57R mutation allows contact with the GDIR motif at the V3 base and that the broader lineage member emerges by eliminating D115 (CDRH3 c-terminus) self-contact with Y98 at the end of heavy chain framework 3 by R98T mutation. Both of these key mutations are rare AID mutations and may therefore represent a barrier to induction of these bnAbs. The authors conclude by suggesting that V3 glycan constructs used for the co-crystals might be useful immunogens.

Overall, the paper is succinct, clear and well written. A few comments and largely minor criticisms are as follows:

**Fig 5C is a nice summary of the findings, but it would be better if framework and CDRs were included for orientation purposes*

The framework and CDRs are now indicated in Fig. 5C.

**A paper by Sok-Burton in Immunity 2016 PGDM12, 14, 21 targeting the high mannose patch GDIR region could probably be cited. Are their features similar to DH270 lineage? Similarly, there are bnAbs from the VRC reported in JVirolgy Longo et al. Dec 2016. There may be others and they should be cited as a way to mine similarities and differences as may be relevant for vaccine design. Possibly these could be aligned as in Fig 5C to show similarities and differences to assist in the narrative in the discussion (page 12 line 251) and help make the point regarding differing CDR lengths mentioned. Even if this information is present in a preceding paper, it is arguably worth some extra space here to emphasize the point.*

We have included sequence alignments with the bnAbs targeting the high mannose patch reported in Sok et al. (2016), Longo et al. (2016), and MacLeod et al. (2016), in addition to PGT124, PGT128 and DH270 lineage bnAbs, in Supplementary Figure 8. The arrows in our figure point to the improbable mutations we discuss in our study, in addition to the conserved D115 residue of the antibody heavy chain, which is distracted by an R98 from forming a bidentate interaction with the HIV Env glycan. Position 57 of the heavy chain is variable, but position 98 is frequently an Arg. We point out these features in the “Discussion” section of the manuscript.

We also did note that some of the antibodies reported in MacLeod et al. (2016) have comparably short CDRs, similar to those of the DH270 lineage, and have low levels of autoreactivity, thus corroborating our claim that it may be possible to elicit bnAbs that are neither auto- nor poly-

reactive. Furthermore, we did note that Sok et al. (2016) identified bnAbs that target the D1 arm mannose of Env, like the DH270 lineage members we describe. These points are now included in our “Discussion” section.

**I could not fully understand Fig. 3. Some labeling of residues might help to orient the reader. The same goes for Supp figures 3 and 4*

The glycans, GDIR motif, and CDRH3 loops are now labeled in Figure 3. The individual monosaccharides in the glycan in Supp. Figure 3, along with individual amino acid residues in Supplementary Figures 1 and 7 (which used to be Supplementary Figures 3 and 4) are also labeled.

**I was not clear on the meaning of the NMR data. Can the interpretations be explained a bit more explicitly?*

To make the NMR data more clear, we have now labeled the peaks in the spectrum that is provided in Supplementary Figure 3 and also included the structure of the Man₉-V3 glycopeptide with the methyl-contributing residues highlighted in that same figure.

Man₉-V3 has eight residues with methyl groups that could contribute to the peaks we observed in the NMR spectrum (5 Ile, 2 Thr, and 1 Ala). This suggests 13 possible peaks, but we observe only 11 peaks, probably because some peaks are overlapping with one another. The isoleucines in the peptide are found in chemically distinct environments (in the β-hairpin region, as well as in the GDIR bulge) and so they are expected to have different peaks in the spectrum. Since only 11 peaks are observed, we can confidently say that at least four of the isoleucines contribute their methyl groups to 8 of those peaks (the other 3 peaks could derive from Ile, Thr, and/or Ala residues).

We have expanded our explanation in the “Results” section to indicate these points.

**I also felt that a sentence or two could be added to explain why the key mutations (above) are cold spots. Why is this so? References are given, but arguably the story is incomplete without at least some brief explanation as to why cytidine deaminase makes some mutations more difficult to achieve than others.*

We included a couple of sentences explaining that somatic hypermutation is a biased process by which activation-induced cytidine deaminase (AID) shows preference towards certain micro-sequence motifs. The G57R mutation we mention occurs at a cold spot sequence motif (GRS, where R is a purine and S is either a C or a G) and the underlined residue is mutated to a C.

**line 247 “Tyr48L is actually Leu48 in the UCA and DH270.3 and is not shown in fig 5B or mentioned in results and maybe it should be.?”*

Leu48 is now indicated in figure 5B and it is also mentioned in the “Results” section.

**a mutation of R98T in DH270.3 might be a nice complement to the KO mutation done in DH270.6 to make the point more concrete.*

We introduced the R98T mutation in DH270.3, but this Fab mutant did not express well and so we were unable to test its binding to HIV Env. It is possible that this mutation introduced instability in DH270.3. This may explain why the R98T mutation only appeared in the branch leading to DH270.6 (the reduced stability from R98T may have been counteracted by other stabilizing mutations), and once it appeared, remained fixed in the branch leading to DH270.6.

**It's not immediately clear how the V3 glycan construct could be used as an immunogen and in fact it was used in ref 17 and did not induce any Nab responses. It does seem reasonable to suggest that it could be useful in a prime boost format. One does wonder if sufficient helper epitopes would be present to make it an effective immunogen without e.g. PADRE.*

We have clarified in the “Discussion” section that the V3-glycopeptide can be used in a prime-boost format with multimerized V3-glycopeptide and/or Env trimers or in conjunction with a T-helper glycoconjugate, such as PADRE. The V3-glycopeptide was in fact used in the Alam et al. (2017) reference but a limitation of that study was that the immunization only used the monomeric form of the V3-glycopeptide, which did not have T helper epitopes and was not very immunogenic. The aforementioned formats should improve the antibody responses from immunization with the glycopeptide.

Reviewer #2 (Remarks to the Author):

The authors describe the structure of a broadly neutralizing HIV-1 antibody, DH270.6, in complex with its glycopeptide epitope as well as the structure of a less broad intermediate clonal family member in complex with a glycan. The structures reveal various details on evolution of the antibody lineage and confirm that the epitope of the antibody can be recapitulated by a synthetic glycopeptide. Both findings are of interest to the field. The authors previously published a structure of the a member of the DH270 lineage as well as the lineage itself, but the current structures are at higher resolution and the presence of the glycopeptide allows for a more detailed structural interpretation of the affinity maturation pathways.

Specific comments:

1. Discussion. Lines 230-231. The authors imply that the glycopeptide in solution assumes the same structure of as this region in the Env protein. However, the glycopeptide might very well attain this particular structure using the antibody as the template (induced fit). Considering the central role of the glycopeptide in this manuscript it is important to flesh out this subject a bit more. The NMR data are OK, but not very convincing to imply that the peptide is conformationally stable in solution. Additional in solution assays such as circular dichroism or other could provide more information on the structure of the glycan peptide in solution. Isothermal titration calorimetry could be used to assess whether antibody binding is accompanied by substantial conformational changes, and/or increase in order.

To supplement the NMR data we originally reported, we used biolayer interferometry to explore the binding kinetics of the glycopeptide with the DH270.6 Fab. We found that the glycopeptide binds with a similar association rate and dissociation constant as the gp120 monomer to the Fab, suggesting that that glycopeptide does not undergo a slow folding step before associating with the Fab. The equilibrium constants for the two complexes were also similar, suggesting that the conformation of the unbound peptide resembled that of the corresponding segment in intact gp120. These data are now presented in the “Results” section of the manuscript, and highlighted in Table 1 and Supplementary Figure 4.

2. Furthermore, the impact of the manuscript could be enhanced substantially by using the structure to design more conformationally stable versions of the glycopeptide.

The structure we determined does suggest potential ways to design a modified Man₉-V3 with increased stability. For example, H330 could be replaced with a residue, such as glutamine, which could potentially hydrogen bond with Thr297, to further stabilize the hairpin. We included this in the “Discussion” section of the manuscript and highlight these residues in Supplementary Figure 3C.

3. Results. In order to make more general statements on V3 broadly neutralizing antibody recognition, it would be good to include figures comparing the structures of DH270.3, DH270.6 and other antibodies targeting this region such as PGT124 and PGT128. Does the 332 glycan assume the same structure in all complexes? And the 301 glycan? It would be good to show this visually. Similarly for the peptide component (i.e. GDIR). It would probably be appropriate to add such figure to Fig. 3. The authors could generate an overlay of the individual glycans and GDIR for DH270.3, DH270.6, PGT124, PGT128 and possibly others and compare the contact residues in epitope and paratope.

We have included overlays in Figure 3 of the GDIR region derived from multiple Env-Fab complexes (from various HIV strains), as well as unliganded Env, to show that the V3 region assumes a similar conformation in all complexes. The glycans did not assume the same structure in the complexes when comparing DH270 lineage members to those of other bnAbs, as well as when comparing other bnAb-complexes to one another. We believe this is due to the conformational flexibility of these long carbohydrate chains, which allows the antibody to bind to the glycan component of the HIV envelope with an “induced” fit. This is now stated in the “Results” section, and shown in Supplementary Figure 6. The V3 region polypeptide chain, on

the other hand, is likely more rigid than the glycans, which is why its conformation appears unaffected by antibody binding.

We also now include images of the PGT124- and PGT128- Env complexes to illustrate similarities and differences to the DH270-Env complex that we discuss in the “Results” and “Discussion” sections. Specifically, we highlight contacts made with the D1 arm of the mannose moiety of the N332 glycan, and the GDIR motif of the V3 region.

4. Introduction. About half of the introduction is a summary of results and conclusions, not introduction. The actual introduction is extremely short.

We have added additional background information in the “Introduction”, specifically on the glycans targeted by broadly neutralizing antibodies of the type we describe. We also discuss in greater depth HIV Env glycosylation, to address the point raised in statement 5 below. Finally, we added some background information on the Man₉-V3 glycopeptide, which was reported in other studies.

5. Introduction. Lines 59-60. Most glycans are indeed heterogeneous, but it appears from studies from the Crispin and Paulson labs, that the HIV Env glycans are more homogeneous than those on other glycoproteins. Glycans at many positions on Env are found exclusively as oligomannose, or even only Man₉ glycans.

We have revised our “Introduction” section and clarified that the glycan sites on HIV Env that are targeted by broadly neutralizing antibodies are of the high-mannose type. We referenced the work done by the Crispin and Paulson labs on this subject. We also explain that different Env constructs may not have the same steric environment as seen by the oligomannose patches on Env, and thus may have different glycosylation patterns, which is why chemical synthesis of glycopeptides would be very useful for vaccine design.

Fig. 2B. The size of the glycans in the left part of the figure is not proportional to the size of the peptide. I would suggest to size glycans and peptide proportionally and remove the right part of the figure (but transfer the labels to the glycans in the left side of the figure).

The glycans in Figure 2B have been resized to be proportional to the size of the peptide and the glycan labels have been moved as suggested.

REVIEWERS' COMMENTS:

Reviewer #2 (Remarks to the Author):

Although I would have wished to see additional in solution data on the conformation of the glycopeptide, as well as structure-based improvement of the peptide, the authors have addressed most of my concerns adequately.